# Profiling the Typical Training Load of a Law Enforcement Recruit Class

**DOI:** 10.3390/ijerph192013457

**Published:** 2022-10-18

**Authors:** Danny Maupin, Ben Schram, Elisa F. D. Canetti, Joseph M. Dulla, J. Jay Dawes, Robert G. Lockie, Robin M. Orr

**Affiliations:** 1Faculty of Health Sciences and Medicine, Bond University, Gold Coast, QLD 4226, Australia; 2Tactical Research Unit, Bond University, Gold Coast, QLD 4226, Australia; 3School of Kinesiology, Oklahoma States University, Stillwater, OK 74078, USA; 4Department of Kinesiology, California State University, Fullerton, CA 92831, USA

**Keywords:** academy, tactical, conditioning, police, cadet

## Abstract

Law enforcement academies, designed to prepare recruits for their prospective career, represent periods of high physical and mental stress, potentially contributing to recruits’ injuries. Managing stress via monitoring training loads may mitigate injuries while ensuring adequate preparation. However, it is vital to first understand an academy’s typical training load. The aim of this study was to profile the typical training load of law enforcement recruits over the course of 22 weeks. Data were prospectively collected using global positioning system (GPS) units placed on recruits during a portion of the academy training, while a desktop analysis was retrospectively applied to six other classes. A Bland–Altman plot was conducted to assess the agreement between the two methods. A linear mixed model was conducted to analyse the difference in distances covered per week, while other variables were presented graphically. Adequate agreement between the desktop analysis and GPS units was observed. Significant differences (*p*-value < 0.01) in distance covered (9.64 to 11.65 km) exist between weeks during early academy stages, which coincide with increases (~6 h) in physical training. Significant decreases in distances were experienced during the last five weeks of academy training. Most acute:chronic workload ratios stayed between the proposed 0.8 to 1.3 optimal range. Results from this study indicate that large increases in training occur early in the academy, potentially influencing injuries. Utilizing a desktop analysis is a pragmatic and reliable approach for instructors to measure load.

## 1. Introduction

Law enforcement is a predominantly sedentary occupation by nature that is interspersed with periods of high-intensity activity [1]. Occupational tasks can range from performing deskwork to near-maximal physical exertions [2]. These shifts from sedentary periods to bouts of high-intensity work can often occur without warning, hindering the ability of officers to properly prepare themselves [3]. These activities may also be performed while carrying approximately 10 kg of equipment (i.e., body armour, handcuffs, and a firearm) [4] which negatively impacts the officer’s capabilities [5]. The physical demands of this career predispose officers to high rates of injury and require adequate fitness to effectively perform their unique occupational tasks.

A critical review by Lyons et al. [6] found that law enforcement officers tend to suffer injuries at a rate of 240 to 2500 injuries per 1000 personnel per annum. For comparison, a study of United States construction workers found an average annual injury rate of 22.10 per 1000 workers [7]. These injuries can impact individual officers by causing decreases in occupational performance [8]. Due to law enforcement’s role in protecting the public, decreases in performance can potentially lead to public harm or even death. At an organizational level, injuries lead to higher financial costs, as seen in an annual report from the New South Wales Police Force showing the median cost per injury claim was AUD 9900 [9]. This same report also found that the average amount of time lost per employee due to workplace injury was 78 h [9]. This absenteeism further increases workforce strain by requiring other law enforcement officers to complete additional shifts to cover the missed time of their injured peers. This could then increase their risk of injury through additional workplace exposure [10]. As such, it is not surprising that multiple methods have been trialled during academy training to reduce injuries, such as movement screening [11] and ability-based training [12].

In addition to lowering an officer’s chance of injury [13], increasing physical fitness can improve occupational performance [14]. Research conducted by Canetti et al. [14] found a significant positive relationship between measures of anaerobic strength, such as muscular strength, and the ability to perform occupational tasks (e.g., body drags or fence climbs). Furthermore, research by Lockie et al. [15] reports that performance on pull-ups, push-ups, and 2.4 km runs were predictive of performance in certain occupational tasks, such as fence climbs and occupational task-based obstacle courses. Recruits who have higher levels of aerobic fitness are also more likely to successfully complete academy training [16]. Furthermore, increased physical fitness has also been found to contribute to improved long-term health outcomes for both the physical [17] and mental [18] health of officers.

Strategies aimed at reducing injury risk while improving fitness are of vital importance to this population, positively benefiting both organizations and individuals. Training academies, due to their safe, supervised, and controlled environment, are an ideal environment to implement such strategies when compared to the unpredictable nature of the occupation [19]. One potential methodology to be researched in academies and to impact both injury risk and fitness is the optimization of training load (TL). Though indirectly affected through concepts such as ability-based training, the optimization of TL has not been studied in depth within tactical populations despite success in the sporting realm [20].

The optimization of TL has recently grown in popularity in the sporting world and is used as a strategy to decrease injury risk while improving fitness and performance [21]. This approach employs a wide variety of tools to measure TL [21]. Approaches to measuring TL can be organized into external and internal loads, and in sports are typically measured via technology such as global positioning system (GPS) units or subjective wellness questionnaires [20,21,22]. Measures of external loads include variables such as distance runs, the volume of weight lifted, or the number of accelerations, while internal loads include variables such as heart rate or ratings of perceived exertion (RPE) [20,21]. Rapid changes in external or internal loads may be indicative of future injury risk or performance change. For example, Piggott [23] found that 40% of injuries in Australian football followed a change in the TL in the previous week. The acute:chronic workload ratio (ACWR) is one approach designed to capture an individual’s cumulative load by comparing their most recent TL (acute) to their prior TL (chronic) [20]. The ACWR has been related to injury risk in the literature [24,25], with ratios of 0.80–1.30 theoretically resulting in the lowest risk of injury [20]. The ACWR quickly gained popularity in sports and has been used across a variety of research studies [24,26,27]. Despite this initial popularity, recent research has suggested that the ACWR may not predict injury due to conceptual faults [28] and methodological flaws [29]. Although the ACWR may not be an effective means of predicting injury risk, it may be able to be used to provide a snapshot of workload over time through which to inform overall TL and progression. Optimizing TL may prove to be a beneficial strategy to reduce injuries and improve fitness in a law enforcement recruit population. Prior to making recommendations, however, it is vital to first profile typical TLs experienced by recruits. Therefore, the aim of this study was to profile the typical TL of a law enforcement recruit class undergoing academy training.

## 2. Materials and Methods

### 2.1. Subjects

Training and schedule data were provided from seven recruit classes from one United States law enforcement agency. All classes took place in the same location but under the supervision of various staff members. Course length did differ between classes, with one class lasting 20 weeks, and the six retrospective classes being 22 weeks in length. GPS data were prospectively collected from a subsample of 24 recruits, 9 female (age = 29.9 ± 6.4 years, height = 163.4 ± 6.5 cm, body mass = 68.2 ± 11.2 kg) and 15 male (age = 35.5 ± 11.9 years, height = 176.1 ± 9.8 cm, body mass = 82.6 ± 11.9 kg) recruits, randomly selected from one class. Informed consent was provided by the recruits and ethical approval was given by the Bond University Human Research Ethics Committee and by the California State Fullerton Institutional Review Board under HSR-17-18-370.

### 2.2. Procedures

#### 2.2.1. GPS and Desktop Analysis

This study profiled the TL of seven classes undergoing academy training. In order to gain an understanding of the typical TL within this population, the first class was profiled using data collected via Polar Team Pro sensors (Polar Electro Inc. Bethpage, New York, NY, USA) collected for four weeks (19 days) in an academy training facility in the United States. Academy training typically lasts 20 to 22 weeks at this facility and consists of classroom lectures as well as physical and skills training sessions. Training typically occurred five days a week, 8 h a day for a total of 40 h of training per week.

Using these data and information obtained from the academy staff, a desktop analysis of the remaining 16 weeks was performed. The desktop analysis was constructed from course schedules and physical training descriptions collected as part of the standard operating procedure provided by academy staff. The TLs of the remaining six classes were retrospectively assessed using this same methodology. The desktop analysis method has previously been utilized in research, albeit in a military population [30].

The Polar Team Pro sensors (Polar Electro Inc. Bethpage, New York, United States) used to measure distance in this study were reported to have less than 5% error when measuring total distance at various speeds [31]. Sensors were applied upon entry to the locker room first thing in the morning and removed upon leaving the classroom at the end of each working day. The distance that recruits covered to and from their parking lot was not collected. Due to the limitations in the data collection process, such as being able to use the Polar Team Pro sensors for only four weeks, total distance was the only variable used for comparison in this study.

For the desktop analysis conducted on the seven classes, all estimates were based on a cohort rather than an individual level. Outliers who did not participate in specific activities were excluded from that specific activity. In situations where the class split into multiple groups, one group was followed and analysed. While this procedure may affect the timings of the load experienced by recruits, the overall load experienced should be similar. This protocol has been used in previous research investigating military recruit training [30]. Distances covered for the desktop analysis were either measured on-site, or retrospectively using the Google Maps measuring tools, and accounted for the distance that recruits needed to cover to reach their respective parking lots. Total distances per week were calculated as well as weekly changes in the distance covered. This was calculated by subtracting the current week’s total distance from the total distance of the preceding week. Data analyses were performed separately to the collection and analysis of the Polar Team Pro (Polar Electro Inc. Bethpage, New York, United States) data to ensure minimal risk of bias.

#### 2.2.2. Calculation of ACWR

Upon completion of the desktop analysis both the ACWR, based on rolling averages, as well the EWMA ACWR were calculated and compared against the proposed 0.80 to 1.30 optimal range [20]. The rolling average ACWR was calculated using a 1:4 week acute:chronic ratio, with one week representing the acute workload and four weeks representing the chronic ratio. Calculating the EWMA ACWR for a given day has previously been described by Williams et al. [32] and is presented in Equation (1).

Equation (1). Exponentially Weighted Moving Average Calculation
(1)$${\mathrm{EWMA}}_{\mathrm{TODAY}}={\mathrm{Load}}_{\mathrm{TODAY}}\;\times \;\lambda a+(1-\lambda a)\;\times \;{\mathrm{EWMA}}_{\mathrm{YESTERDAY}}$$
where λa is between 0 and 1 and represents the degree of decay. Higher values of λa will discount older observations at a faster rate. The calculation of λa can be seen in Equation (2).

Equation (2). Degree of Decay Calculation
λa = 2/(N + 1)(2)
where N is the chosen time decay constant.

The same one-week acute workload and four-week chronic workload were used for the EWMA ACWR leading to an N value of 7 days for the acute workload and 28 days for the chronic workload. To calculate the EWMA ACWR value itself, an EWMA was calculated for both the acute and chronic workload using the above formulas. The EWMA ACWR value was then calculated by dividing the EWMA acute workload by the EWMA chronic workload. To begin the calculation, the first observation in the series was arbitrarily recorded as the first workload in the series.

#### 2.2.3. Physical Training Modalities

Times spent training or completing various activities were calculated and split into the following categories: anaerobic, aerobic, muscular conditioning, multi-modal, classroom, and skills training. This analysis was completed via a desktop analysis and informed by the lead author based on time spent on location and expertise of the training staff. Anaerobic training was defined as high intensity, intermittent bouts of training and aerobic training as of low intensity, but long duration. Muscular conditioning was defined as periods of training that emphasized weightlifting and other activities focusing on increasing muscular strength, while multi-modal training was activities that encompassed a combination of some or all of the aerobic, anaerobic, and muscular conditioning, such as circuit training. Skills training encompassed a variety of activities, such as weapons handling, evasive driving, or practising real-world scenarios (e.g., pulling over a suspect), while time spent in class was any time spent receiving a lecture (Table 1). Generally, on any day, recruits participated in a mixture of classes, skills training, and physical training sessions. Physical training sessions generally occurred two to four times a week and lasted for one to two hours in duration with programs varying between classes, though this varied with some weeks not containing any planned physical training [33].

### 2.3. Statistical Analysis

Data are reported as mean ± standard deviation (SD) unless otherwise specified. Descriptive methods of data normality were completed (i.e., distribution plots, skewness, kurtosis, and outliers) before analysis to determine the appropriateness of parametric or non-parametric analyses. A Bland–Altman plot was produced to determine the agreeability of the desktop analysis against the mean distance data from the Polar Team Pro sensors. A Bland–Altman was chosen due to its improved ability to demonstrate agreement compared to correlation [34,35]. The Bland–Altman allows for the assessment of bias (encompassing fixed, proportional, and systematic bias) and precision (the closeness of the limits of agreement as measured by +/−1.96 SD) [34,35]. Assumptions (normal distribution of differences, no correlation between differences and means, and heteroscedasticity) were checked and met.

The Bland–Altman was conducted utilising only data captured during outdoor training sessions. This was selected due to a decreased indoor accuracy from the Polar Team Pro sensors and the potential for the signal to be dropped and picked up in a different location while the recruits were sitting in class (‘geographical drift’). Therefore, the points of comparison for the Bland–Altman plot were the mean distance covered by participants wearing the Polar Heart Rate Monitor, and the expected distance covered during these same sessions as identified by the desktop analysis. This resulted in nine data points from the sensors for the production of the Bland–Altman plot. It was expected that the Polar Team Pro sensors would consistently demonstrate higher measures of distance than the desktop analysis due to the sensors being able to gather data on incidental distances covered. Absolute percentage error (absolute mean difference/mean) was also calculated for the means of both the Polar Team Pro sensors and the Bland–Altman plot.

Due to the differing class lengths, a linear mixed effects model was conducted to analyse the distance covered across various weeks. This was accomplished by using weeks as a predictor of distance and specifying weeks as a random effect nested within separate classes [36]. Post hoc analysis was conducted using Tukey’s method to correct for multiple comparisons. This comparison was completed across all weeks, with corrections for *p*-values performed for all iterations. To make the results more practical, comparisons are only shown between six adjacent weeks. For example, week one was compared to weeks two, three, four, five, six, and seven. This timeframe was chosen based in part on periodization principles outlined by Issurin [37]. Significance was set at *p*-value < 0.05 a priori. All statistical analyses were conducted using RStudio (RStudio, Public Benefit Corporation, version 1.25.042) with packages blandr, nlme, ggplot2, multcomp, pastecs, and reshape.

## 3. Results

A mean difference of 0.27 km (95% CI 0.20–0.34 km) for the distance covered during outdoor training sessions existed between the Polar Team Pro sensors and desktop analysis, indicating fixed bias was present as the Polar Team Pro sensors consistently measured higher than the desktop analysis, as expected. There does not appear to be a proportional bias as the mean difference was consistent across the range of mean distances covered (Figure 1). This is reflected by a slope of 0.019 when calculating a regression equation between the differences and means of the measures. There is also no systematic bias (correlation between the difference scores and the mean scores) (*r* = 0.38, *p*-value = 0.31). The absolute percentage error was 7.7% and 8.3% of the average estimated distance by the Polar Team Pro sensors and desktop analysis.

Though the timeframes varied between classes, the average distance covered per week during the 20-week program was within one standard deviation of the overall mean (Table 2). This suggests that the difference in distances between the programs was likely mainly due to the difference in duration, and that the seven classes can be reasonably compared together.

### 3.1. Distance Covered

There was an approximately 10 km increase in total distance covered from Week 1 to 2 (Figure 2). Results of the linear mixed effect model show significant differences between Week 1 and Weeks 2 (difference = 9.65 km; *p*-value = <0.01), 3 (difference = 9.64 km; *p*-value = <0.01), 4 (difference = 11.65 km; *p*-value = <0.01), 5 (difference = 9.69 km; *p*-value = <0.01), and 6 (difference = 10.07 km; *p*-value = <0.01). Overall, the model shows that week was a significant predictor of distance (χ^2^(21) = 88.41, *p*-value < 0.001), suggesting that the distance covered changes significantly across weeks.

Between Weeks 7 and 14, the distances covered plateaued, remaining between approximately 15 and 20 km per week. At Week 17, the total distance covered began to decline to less than 10 km during the last two weeks. This decrease was also reflected in the linear mixed effect model, with significant differences between Week 15 and Weeks 18 (−8.96 km; *p*-value = <0.01) and 21 (−13.16 km; *p*-value = <0.01); Week 16 and Weeks 21 (−9.82 km; *p*-value = <0.01) and 22 (−9.69 km; *p*-value = <0.01); and Week 17 and Weeks 18 (−7.99 km; *p*-value = 0.049), 21 (−12.19 km; *p*-value = <0.01), and 22 (−12.07 km; *p*-value = <0.01). For full results of the linear mixed effects model including *p*-values, please refer to Appendix A.

As per the weekly total distance covered, there was an increase of approximately 10 km in distance covered during Week 2 (Figure 3). No other large changes appeared to occur during most of the academy. However, over the last five weeks, larger fluctuations could be seen, ranging from approximately −9 to 4 km changes from week to week.

### 3.2. Acute:Chronic Workload Ratios

The average ACWR showed few spikes and consistently remained between the 0.80–1.30 range through most of the program (Figure 4). The highest ratio of approximately 1.20 occurred during Week 4. From Week 16 on, the ACWR began to decrease, with intermittent increases between Weeks 18 and 20 (0.70 to 1.00), and Weeks 21 and 22 (0.60 to 0.70). Due to the methodology of the ACWR, results cannot be reported until Week 4, but the impact of large, early increases in distance covered could be seen as the acute load is still higher than the rolling average.

While the ACWR is unable to display data during the first four weeks, the EWMA ACWR can begin tracking from Week 2 (Figure 5). The EWMA ACWR shows an increase to approximately 1.40 during the first six weeks. After this period, the EWMA ACWR begins to consistently decline, ending at about 0.90 during Week 22.

### 3.3. Class, Skills Training, and Physical Training

The hours completed in class and skills training are shown in Figure 6, which suggests an inverse relationship between the two training types. Overall, except for Weeks 8 and 18, recruits were more likely to spend time in class than completing skills training. There was a higher proportion of time spent in class, particularly at the beginning and end of the academy, with the middle time period during academy training showing increases in the time spent training various skills.

The hours spent completing various types of physical training each week are presented in Figure 7. An increase of approximately six hours is seen between Week 1 and Week 2. Time spent training remained relatively consistent for the rest of the program, around three to four hours per week, before tapering off during the last five weeks. Aerobic conditioning and multi-modal training were the two most common forms of training, with less time spent on anaerobic and muscular conditioning.

## 4. Discussion

This study profiled the typical TL of a law enforcement recruit class during academy training by analysing the total distance covered (inclusive of weekly changes and ACWR), and hours spent in class, conducting skills training, and completing physical training. The results of this study showed an increase in the total distance covered during the early stages of the academy, especially between Weeks 1 and 2 (approximately 10 km). The effect of this almost twofold increase is also reflected in the higher early values of the ACWR and EWMA ACWR. The data also highlighted that recruits often completed aerobic and multi-modal based training, as opposed to anaerobic and strength-based training, echoing previous findings in this population [38].

Results from the Bland–Altman plot showed that a desktop analysis could be used to gauge the total distance covered by recruits during the academy, though it may underestimate the true distance covered. The tendency for the desktop analysis to underestimate prospective and observed measures has been previously acknowledged in the literature [30]. These differences are likely due to incidental movements that would be registered on a GPS device but not a desktop analysis (e.g., recruits getting up and moving around during a class session) [30].

The desktop analysis of this population showed an increase of 10 km in total distance covered per week after the first week and remained between approximately 15 and 22 km until Week 17. These results are similar to a study by O’Leary et al. [39] who reported weekly distances between 10 and 18 km in British military recruits, as measured using GPS technology. This increase in distance corresponded to an increase in hours spent performing physical training, potentially resulting in a higher training intensity. Results from the linear mixed effect model showed significant differences between Week 1 and Weeks 2–6, with differences ranging from 9.65 to 11.65 km. Previous research in law enforcement populations has suggested that injuries occur, in part, due to the increase in physical demand as recruits begin academy training [40]. In fact, a profile of injuries in this same law enforcement population found that injuries often occur to the lower limb during physical training, with the majority of injuries occurring between Weeks 2 and 5, and Week 2 specifically having the largest injury total [41]. In military academies, which employ similar training strategies as law enforcement [12], research has found a similar trend where the increase in physical training is related to injuries, particularly those that are overuse in nature [40]. Research by Booth et al. [42] has even shown that physiological, biochemical, and psychological evidence of overtraining can be seen as early as in the first five weeks of training in these populations. This overall increase in TL may be a contributing factor to the overtraining and high rates of injuries seen in recruits [40,42]. Though TL may be a contributing factor to the injuries seen in recruits, it is important to note that injuries are the result of a complex interaction of factors such as age, sex, and weight, among others [43].

Previous research on elite Australian Football League (AFL) players has shown that exposure to week-to-week changes of >30% in distance may increase the risk of injury but had poor predictive ability (area under curve = 0.55–0.56) [44]. Additional research on AFL players found that three-weekly cumulative distances between 73.72 and 86.66 km increased the risk of injury (OR = 5.49; 95% CI = 1.57–19.16) [22]. In this study’s population, week-to-week changes of over 30% were recorded, particularly from Week 1 to Week 2 where distances covered almost doubled, as well as large three-weekly cumulative distances (Weeks 2–4 showed approximately 60 km in distance covered). These variables may be contributing to injury risk in this recruit population. Caution does need to be applied when comparing a sporting population to law enforcement recruits. The distance covered by athletes, particularly AFL players, may be at a higher average intensity which could contribute more to injuries. Additionally, elite AFL players have previously been found to have a VO_2Max_ of 63.40 mL/kg/min [45], while the average recruit in this population has an estimated VO_2Max_ of 35.42 to 41.08 mL/kg/min [33]. Given the beneficial impact of physical fitness on injury risk in both of these populations [13,46], elite AFL players may be able to tolerate a higher TL than recruits. Due to differences between sporting and law enforcement recruit populations, further research will need to be conducted specifically on recruits to study the relationship between the TLs experienced and associated injuries.

In research conducted by Trank et al. [47], military recruits who ran more than 25 miles (40.23 km) over the course of an eight-week training program had an increased risk of injury, with no resultant improvement in 1.5-mile (2.4 km) performance. Recruits in this current population were likely to undergo training programs with an emphasis on long-distance running [33,38,48]. This type of training may be predisposing recruits to higher rates of injury (due to increased exposure [49]) with little benefit to performance. While cardiovascular fitness is crucial to occupational performance in law enforcement [50], it is unlikely that officers will be required to perform long-distance running as an occupational task [2]. Improving cardiovascular fitness through means that are more closely related to occupational tasks may be of greater benefit to law enforcement officers. The utilisation of high-intensity interval/intermittent training (HIIT) may be an effective way to improve physical fitness with a reduction in distance covered (thus decreasing exposure to injury) that more closely relates to occupational tasks. Research has suggested that HIIT training can significantly improve physical fitness in the general population [51]. Though this may decrease the distance covered, HIIT training may impact injury risk due to higher intensities. The relationship between intensity and injury risk will need to be studied in this population to fully understand how HIIT may impact injury risk.

Due to the nature of the ACWR, calculations could not be made during the first four weeks of the academy when the total distance increased significantly. The impact of this sudden increase was better presented in the EWMA ACWR, which resulted in an increase over the course of the first five weeks above the proposed 1.30 optimal range, suggestive of the large increases seen in the total distance covered. In a population of rugby league athletes, an ACWR using a total distance of less than 0.30 and a two-week average greater than 1.31–1.51 was likely to increase injury risk [25]. Similarly, in a cohort of AFL players, an EWMA ACWR greater than 2.00 was associated with increased injury risk when compared to a ratio between 1.00 and 1.49 [27]. Both ACWRs never reached these values in the studied population, possibly due to the calculation utilizing an average across seven classes. Multiple limitations in the ability of the ACWR to predict injury risk have been discussed in the literature. Common criticisms of the ACWR include no physiological underpinning for the recommended safe range to be applied across a variety of sports and TL variables [28], mathematical coupling (due to the same value being included in both the acute and chronic variables) that can lead to spurious correlations [29], and potentially a high false discovery rate due to research analysing the ACWR by splitting a continuous variable into discrete categories, which is known as discretisation [52]. Other limitations have been proposed in the literature, but a whole review of the ACWR is beyond the scope of this article. Thus, readers are directed to other published works for further information regarding the potential limitations of the ACWR [28,29,52,53]. Practitioners should exercise caution when considering utilising ACWRs to adjust training programs with the intent to reduce injuries.

The results of this study suggested that anaerobic and muscular conditioning were the least utilized forms of physical training, with a large focus on aerobic and multi-modal training. These findings are consistent with previous research in this population which suggests a focus on long-distance runs and body weight circuits [48]. It has been found in law enforcement populations that higher levels of anaerobic fitness led to improved performance in occupational tasks [14,54]. Despite this relationship, academies often engage in physical training programs focused on muscular endurance and aerobic fitness [33,48,55]. This training focus is typically due to competing demands for time from other necessary activities, such as defensive tactics or range sessions [16], as well as lacking the necessary resources and equipment to safely and effectively implement programs aimed at developing anaerobic fitness and strength in a large body of recruits [16]. Recent research has highlighted certain options for law enforcement academies to pursue in order to improve these measures of fitness such as the use of unconventional equipment (e.g., body armour or ammunition cans) and the incorporation of unilateral exercises [12,55].

It should be noted that portions of skills training in this population, such as defensive tactics, do contain periods of anaerobic conditioning. However, an earlier study in this same population has found that over the course of the academy, greater improvements were seen in measures of aerobic fitness and muscular endurance when compared to anaerobic fitness, muscular strength, and muscular power [33]. An increase in anaerobic-based training may lead to further improvements in these fitness components and potentially lead to greater occupational effectiveness upon graduation due to the relationship between these measures and occupational tasks [14,15,54]. There was also a decrease in hours spent completing physical training during the last five weeks of the academy. Aerobic performance has previously been linked to occupational assessments such as fence climbs and obstacle courses [15], while eccentric strength has also been linked to occupational tasks (e.g., grappling and landing from jumps) [56]. Therefore, it is vital to ensure that recruits are receiving enough training stimuli prior to joining the workforce to at least retain the improvements experienced during the academy and ensure adequate levels of occupational performance.

There are limitations present in this paper. First, the Polar Team Pro units were only used during the first four weeks of a 20-week academy as researchers were limited in their ability to spend time on location. While training staff may have been able to facilitate further use of these units, it is unlikely they would have had the time needed to consistently download data from the units, recharge the units, clear storage on the units, and reissue the units. The collection of these data is often a full-time role in sports teams and adding this task to training staff would likely detract from their goal of teaching and training recruits to become effective law enforcement officers. This would be in addition to their regular tasks which require an excess of 40 h per week, tending to a multitude of academy duties. Another option is the utilisation of a research team that is sufficiently equipped to provide the technology and human resources to implement a tracking program. A research team would likely need to be contracted to do so, thus representing a large financial cost to the law enforcement institution. The use of desktop-based analysis, compared to individualized load monitoring, especially on a variable such as distance, may be a more practical application given these current barriers (e.g., number of recruits and resource requirements including time and financial) and the typical training performed, though this strategy may be influenced by bias. The absolute percentage error was within 10% of the two methodologies. This was deemed acceptable by the authors based on population expertise and previous experience validating desktop analyses [30]. Fixed biases that have a greater than 10% error should require further investigation into the desktop analysis. Further research will be necessary to understand the validity of a desktop analysis performed by law enforcement staff compared to other measurements such as GPS devices. This study was also not able to provide information on internal TL through the use of heart rate or perceived stress over the full duration of the academy. Further research will need to be conducted using these variables to improve the information on the TL and intensity experienced by law enforcement recruits and how these may relate to injury risk and physical fitness. Additionally, this study was not able to compare the differences between male and female recruits. Previous research in military training has shown significant differences in internal TL variables between male and female trainees, which may affect the injury risk between the two sexes [39]. Injury data are not presented in this article. This limits the ability to draw further conclusions about the relationship between TL and injuries. Future research is needed to understand TL and injuries in a law enforcement recruit population. Lastly, the findings from this study are based off of one United States law enforcement agency and it cannot be assumed that these results will be applicable to other agencies, either within the United States or internationally.

## 5. Conclusions

The use of a desktop analysis may be an appropriate method to track the TL in law enforcement populations, especially with current barriers to individualised load monitoring. The results of this research showed increases in distances covered of more than 10 km weekly, particularly during the beginning of the academy training period. There was also a concurrent increase in the time spent undergoing physical training of approximately six hours from Week 1 to Week 2. These increases in both physical stress, distance covered, and physical training can contribute to the injury risk of law enforcement recruits. Optimizing the training of recruits, particularly during the early stages, may be a valid method to mitigate this risk of overtraining and injury. Staff members currently working in law enforcement academies should also explore the use of anaerobic and strength training strategies to provide a varied stimulus and increase fitness components that are vital to working in law enforcement. The addition of internal TL measures will be necessary for the future to examine the role that exercise intensity has on injury risk. Prior to analysing the relationship between these factors, it is essential to profile the typical fitness changes and injuries experienced by this population.

## Figures and Tables

**Figure 1 ijerph-19-13457-f001:**
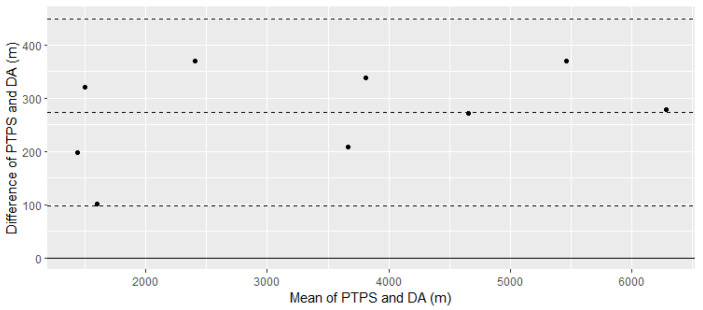
Bland–Altman plot comparing Polar Team Pro sensors and desktop analysis. Key: PTPS, Polar Team Pro sensor; DA, desktop analysis; middle dashed line, mean difference between PTPS and DA; upper dashed line, +1.96 SD from mean difference; lower dashed line, −1.96 SD from mean difference.

**Figure 2 ijerph-19-13457-f002:**
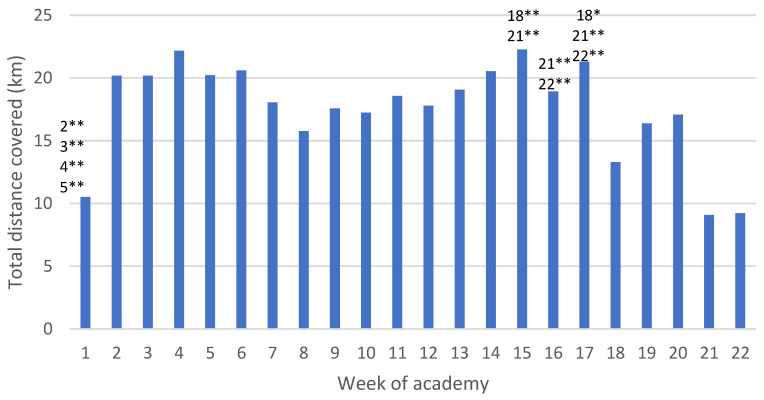
Average total distance covered per week across seven academy classes. Key: Numbers represent the weeks that are significantly different in total distance covered. For example, Week 16 has significantly greater distance covered than Weeks 21 and 22. *: *p*-value < 0.05; ** *p*-value < 0.01.

**Figure 3 ijerph-19-13457-f003:**
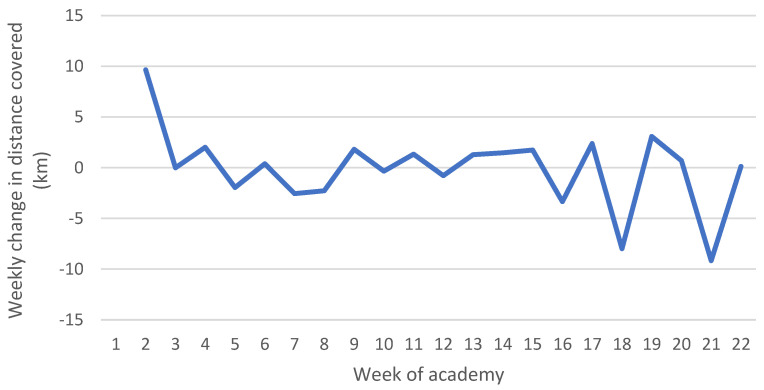
Average weekly change in distance covered across seven academy classes.

**Figure 4 ijerph-19-13457-f004:**
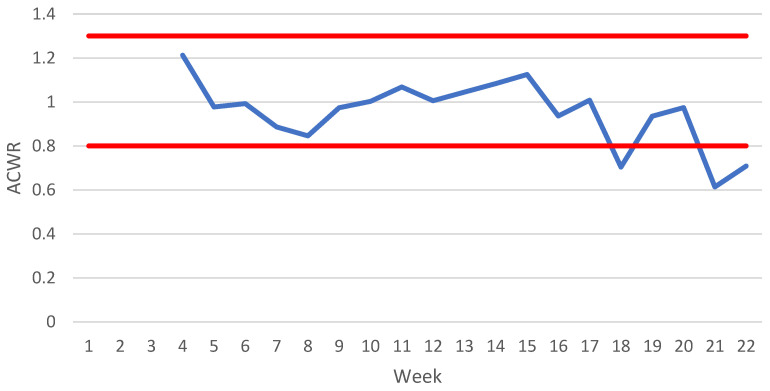
Average ACWR of distance covered across seven academy classes. Key: Red lines signify 0.80–1.30 optimal range as proposed by Gabbett [20]; ACWR, acute:chronic workload ratio.

**Figure 5 ijerph-19-13457-f005:**
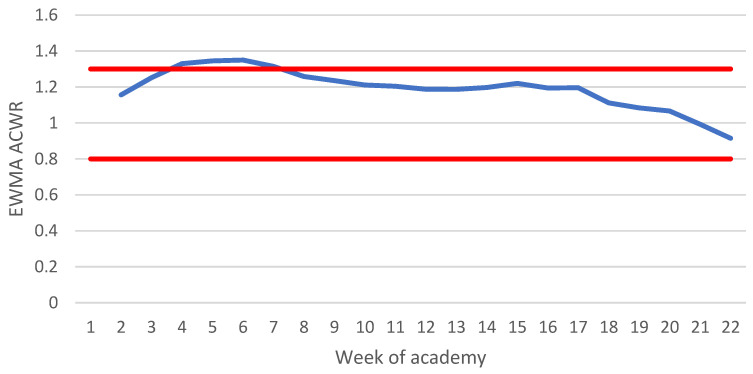
Average EWMA ACWR of distance covered across seven academy classes. Key: Red lines signify 0.80–1.30 optimal range as proposed by Gabbett [20]; EWMA ACWR, exponentially weighted moving average acute:chronic workload ratio.

**Figure 6 ijerph-19-13457-f006:**
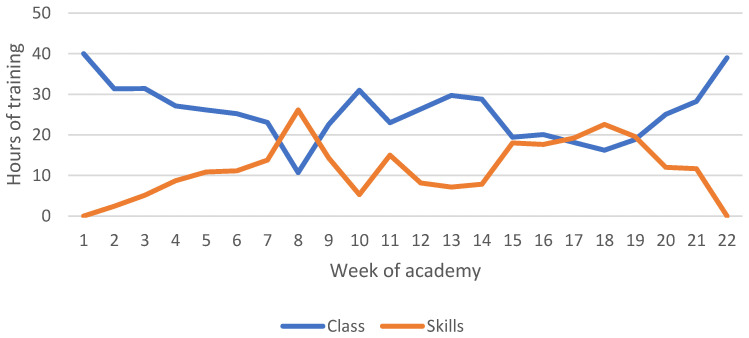
Average hours of class and skill training per week across seven academy classes.

**Figure 7 ijerph-19-13457-f007:**
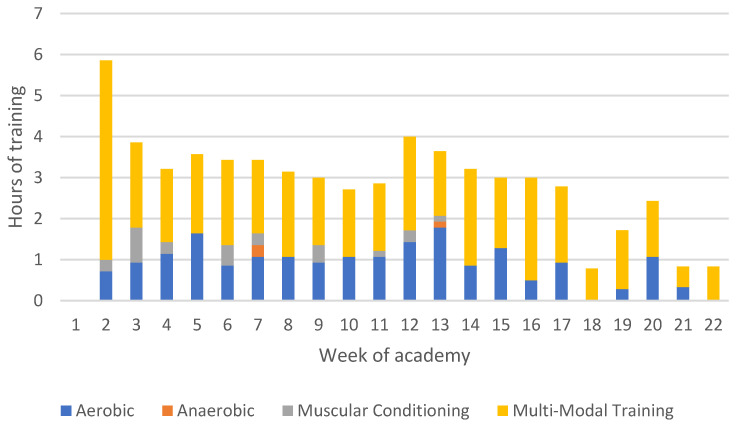
Average hours of training methodology per week across seven academy classes.

**Table 1 ijerph-19-13457-t001:** Training modality definitions.

Training Modality	Definition
Anaerobic	High-intensity, intermittent bouts of training
Aerobic	Low-intensity, long training
Muscular conditioning	Training periods focused on weightlifting and other activities to increase muscular strength
Multi-modal	Training that encompasses a combination of anaerobic, aerobic, and muscular conditioning
Skills training	Training sessions focused on occupational skills such as weapons handling or evasive driving
Classroom	Time spent receiving a lecture

**Table 2 ijerph-19-13457-t002:** Average distance per week per class during academy training.

Class Number and Duration	Average Distance per Week (km)	# of SD from Mean
Class 1—22 weeks	17.99	0.45
Class 2—22 weeks	17.12	−0.68
Class 3—22 weeks	18.72	1.4
Class 4—22 weeks	18.44	1.03
Class 5—22 weeks	16.64	−1.30
Class 6—22 weeks	17.56	−0.11
Class 7—20 weeks	17.04	−0.79
Total mean (SD)	17.64 (0.77)	

Key: # of SD—number of standard deviations.

## Data Availability

Due to the sensitivity of the data, it is available upon reasonable request.

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
