# Peer review of "Profiling the Typical Training Load of a Law Enforcement Recruit Class"

_ijerph, 2022, doi:10.3390/ijerph192013457_

Round 1

Reviewer 1 Report

ABSTRACT 

The Abstract summarizes of profiling the typical training load of law enforcement recruits over the course of  22 weeks. The topic of the article is interesting. However the article has not been well designed and it does not executed to answer the research questions. Some questions could be discussed or clarified to improve the paper. In addition, the submission is not free from mayor editorial errors. 

INTRODUCTION 

The Introduction provided information about what is and what was found about law enforcement and the injury risks in the paper in sufficient details except the methods used to measure the principal variables analyzed.  The authors well explained the scientific background and rationale for the investigation being reported. However, no much researches are shown about different methods to measure the variables analysed. The goals of the study were precisely defined. 

Some minor editorial remarks: 

  •  

METHODOLOGY 

This study analyzed an specific sample which is not representative for a wide range of populations.

Before selecting that methodology would be essential to profile the typical fitness changes and injuries experienced by this population.

How important do you think are these results to science?

RESULTS 

The results has been shown in a clear way. However, please give information on potential confounders that may affect the results. 

Some editorial remarks: 

  • Please sumarise several figures in one. There are too many.
  •  

DISCUSSION 

In the Discussion the Authors summarized key results with reference to study objectives. They also compared their results with other findings. However more studies that compare different methods of measurement are needed.  Discussion was divided into subsections which make this section clear and legible. 

The conclusions are supported by the results.  

The literature is relevant and updated. However, please add literature that araise the methodology used.

Author Response

Please see the attached response. Thank you.

Reviewer 2 Report

The authors present their findings of training load for new law enforcement recruits over a period of a 4-5 months. Using a combination of HR telemetry and geo-tracking, as well as desktop based analysis the authors analyse the total and changes in distance covered week-to-week during the training programs of seven classes, as well as the type of training completed. Their findings suggest that training loads change significantly in the early weeks of training with a sharp increased of 10km for overall distance covered. Additionally, the changes in training load fluctuate throughout the beginning period and final period of the training program. The authors conclude that optimising training loads in the early period of the program to avoid rapid increases in volume, as well as adopting a higher proportion of muscular conditioning and anaerobic style training, may decrease injury risk for new recruits. Additionally, they discuss the underestimation of total distance covered using desktop analysis, even though it may be more practical.

Please see my comments below regarding the manuscript.

General

1. I do not have any significant concerns about the methodology or findings of this manuscript, however the manuscript does require moderate English editing as there are frequent grammatical mistakes throughout. To assist with the correction of these errors, I have highlighted problematic sentences or words in green highlight and attached to my review. There also may be hyphens missing or the tone of the sentence may be passive in nature. Please adjust the grammar, structure and/or spelling of the highlighted sentences. 

Title

1. Please adjust the title so it reads grammatically sound - i.e. either 'Profile of...' or 'Profiling the'

Abstract

1. Please spell out GPS at first use.

Methods

1. Please spell out SD at first usage. 

2. Please provide in brackets manufacturer and location for R software

Discussion

1. Please briefly elaborate on some of the criticisms of ACWR mentioned in line 397. I understand that it may not be appropriate to go into a thorough analysis, but given it is one of the core methods of your analysis, you should at least list the criticisms before referring the reader to the literature. 

2. In your limitations, please briefly discuss a limitation of your study in relation to not collecting data on actual injuries that may have occurred during the period of analysis.

Thank you for your submission and good luck in your research.

Author Response

Please see the attached responses. Thank you for your assistance.

Round 2

Reviewer 1 Report

Thank you very much for your corrections. The manuscript has now the sufficient value to be published.